**Data Availability Statement:** All relevant data are within the manuscript and its Supporting Information files.

# Feasibility and technique of endovenous laser ablation (EVLA) of recurrent varicose veins deriving from the sapheno-femoral junction—A case series of 35 consecutive procedures

**Lars Müller***, Jens Alm

Department of Vascular Surgery, Dermatologikum Hamburg, Hamburg, Germany

* L.mueller@dermatologikum.de

## Abstract

### Aim

To assess the feasibility and technical success of endovenous laser ablation (EVLA) of recurrent varicose veins arising from the former sapheno-femoral junction (SFJ).

### Methods

We retrospectively analyzed all EVLA procedures treated in our institution by one surgeon between March 2019 and April 2020 and selected all consecutive cases with SFJ recurrence occuring after surgical high ligation and stripping or endovenous thermal ablation for incompetence of the great saphenous vein (GSV) in superficial venous insufficiency. The feasibility, technical success as determined by duplex ultrasound on the postoperative visit, complications and rate of endothermal heat-induced thrombosis (EHIT) were recorded. A subgroup definition was performed based on sonographic morphology of the recurrence and resulting strategy of ablation.

### Results

Thirty-five limbs with SFJ recurrence in 34 patients were treated with EVLA in order to shut down the highest refluxing point. In 22 interventions, it was required to switch off a short stump or a neovascularization by direct puncture (Subgroup 1). In 13 treatments, the presence of residual GSV segments, or persistent, varicose transformed major tributaries like the anterior accessory great saphenous vein, enabled cannulation and advancing the laser fiber from distal to the former SFJ (Subgroup 2). The EVLA procedures could be successfully carried out in all 35 cases. There were no major complications, no thromboembolism or EHIT, and no local groin complications. In one case, the patient developed a phlebitic response that required temporary medication. Technical success was achieved with 34/35 treatments (97.1%). When comparing the subgroups, the morphological pattern of the SFJ

**Funding:** The author(s) received no specific Funding for this work.

**Competing interests:** The authors have declared that no competing interests exist.

recurrence and the resulting technique of puncture, cannulation and ablation did not influence the result.

## Conclusions

The results of this case series suggest that SFJ recurrences can also be successfully treated in situations where there are tortuous or short stumps that require direct puncture and ablation.

## Introduction

Recurrences after varicose vein surgery are a common phenomenon. Depending on the length of the observation period, recurrent varices are observed in up to 62% of patients after 11 years of varicose vein surgery [1–3]. Possible sources are refluxing points or vein segments remaining after pre-treatment, which may occasionally result from technical or tactical errors [4–6]. In addition, variceal recurrences may arise in the form of thin-walled, serpentine and sometimes large-caliber veins. These, mainly due to the surrounding scarring caused by the previous operation, are often more difficult to treat surgically, since they can tear and bleed quite easily. The underlying hemodynamic, cellular and molecular mechanisms for the development of such recurrences are poorly understood [7–9]. Some authors believe that the origin of such veins, which are generally referred to as neovascularizations, is the result of the actual formation of new vessels that would justify the term [10–12]. In contrast, there are immunohistochemical indications of a transformation of preformed veins, for example mediated by hemodynamic influences, in the course of the formation of a recurrence [7].

So far, no generally applicable standards exist for the mode of treatment of recurrent varicose veins. In the treatment of primary varicose veins, endothermal procedures are already considered to be the first choice for treatment in some national guidelines, for example in the USA and the United Kingdom, since the burden is lower compared to open surgery with comparable effectiveness [13,14]. However, there is only limited experience with laser or radiofrequency for the treatment of relapses [15–17].

Groin recurrences, which often derive from the former sapheno-femoral junction (SFJ) after varicose vein surgery or endovenous ablation for incompetence of the great saphenous vein (GSV), are of particular clinical relevance. From a surgical point of view, the removal or elimination of recurrent varicose veins at the upper refluxing point is the most consistent form of treatment. In analogy to the high ligation or so-called crossectomy, the recurrence is directly switched off in the area of the SFJ, at the transition to the femoral vein [18]. However, due to the pre-treatment in the same region, the open surgical treatment of such a finding is technically much more complicated than the treatment of primary varicosis and may be associated with significant morbidity [19,20].

In this case series, we evaluated our own, consecutive endothermal procedures over one year and selected those with endovascular laser ablation (EVLA) performed for SFJ recurrence. The primary endpoint was the technical success rate determined during the postoperative control, and the feasibility of the treatment and the complication rate. In addition, a subgroup analysis should be used to compare cases in which treatment was more demanding due to the morphology of the recurrence with those in which the treatment was similar to EVLA for primary varicose veins.

## Methods

### Study population

The electronic medical records of all consecutive patients in whom routine endothermal ablations were performed by a single surgeon (LM) between March 2019 and April 2020 were analyzed retrospectively. According to the classification of recurrent varicosities after surgery (REVAS), all treatments for *same-site* recurrences from the SFJ were selected [4]. Patients with primary varicose disease, or *new-site* and *different-site* varicosities, according to REVAS, were excluded. From treatment cases fulfilling the inclusion criteria, the relevant data were collected in entirely anonymized fashion and stored in an anonymous database (S1 Table). The STROBE guidelines (Strengthening Reporting on Observational Studies in Epidemiology) were employed to review reporting in this study [21].

### Ethics statement

The study was conducted in accordance with the Declaration of Helsinki. After applying for authorisation from the Ethics Committee of the Physician Chamber of Hamburg, the latter determined that local legislation exempts this retrospective analysis of fully anonymised data from the need for ethical approval and informed consent (file number PV7252).

### Preoperative assessment

The assessment and indication for the operation were made using duplex sonography in the standing position using a Logiq P6 Pro (GE Healthcare, Chicago, IL). The indication was established in a symptomatic superficial vein reflux caused by an SFJ recurrence. The anatomy of the recurrence was further examined for diameter, the exact source of reflux and the morphology of the recurrence (Fig 1). Based on the sono-morphological pattern, we assessed whether neovascularization was involved in the genesis of the SFJ recurrence. On the one hand, neovascularization with tangled and tortuous veins can directly adjoin the femoral vein and mediate reflux. However, it can also be connected downstream of an SFJ stump, or between a stump and subsequent straight vein segments. In the presence of a relatively straight segment with reflux, e.g. a GSV remnant, or a large side branch, these veins were intended to undergo concomitant laser ablation. Furthermore, the medical history was recorded with all relevant demographic and medical parameters.

### Surgical technique

All work steps were carried out under sterile conditions and permanent ultrasound guidance (Fig 2). The choice of anesthesia method was dependent on the patient's wishes, but we recommend general anesthesia when treating short stumps. For peri-venous tumescence, physiologic saline solution was utilized in case of general anaesthesia. Without the use of systemic narcotics, local tumescent anesthesia (1000 mL physiological saline + 50 mL Mepivacaine 1% + 8 mL sodium bicarbonate 8.4%) was applied. A portable ultrasound system (Logiq e, GE Healthcare, Chicago, IL) was utilized for the intraoperative ultrasound. The ablations were employed with 1470nm 2-ring radial fibers (ELVeS® Radial®, Biolitec AG, Jena, Germany), either the 1.8 mm fiber for larger diameters or the 1.3 mm fiber. In two cases, we made the ablations with a 1.3 mm 1940nm radial fiber (iMS Saturn Slight Fiber IRH 400, iMS GmbH, Tutzing, Germany). The procedure for ablation was based on the morphology of the SFJ recurrence. If the vein area to be treated at the former SFJ was a short stump, a meandering vein segment or neovascularization, this proximal vein section was punctured directly. The insertion instruments

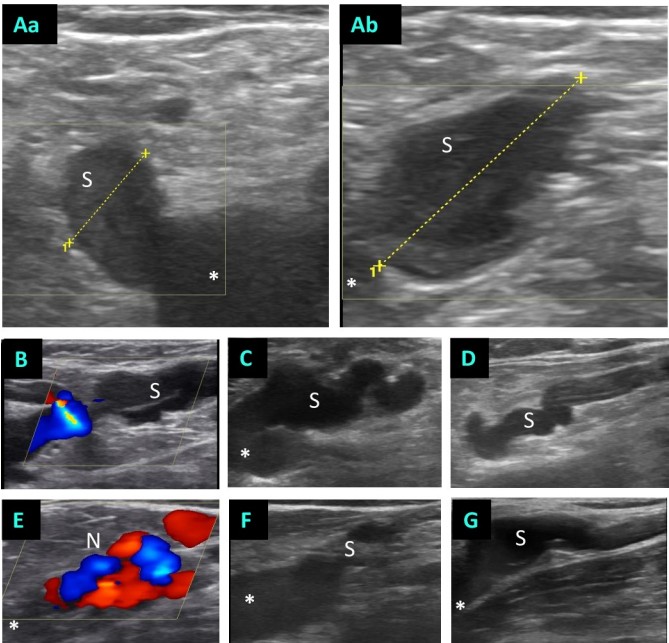

**Fig 1. Morphology of SFJ recurrences.** Representative duplex ultrasound images. (A) measurement of the stump of a left-sided SFJ recurrence. (Aa) measurement of the diameter in cross-section; (Ab) Measurement of the length of the stump in the longitudinal plane. (B-D) Examples of SFJ recurrences with axial displacement. (E) An example of neovascularization in the area of the SFJ. (F, G) Stumps which is connected to a relatively straight vein segment with only limited axial offset. * = femoral vein; S = stump, N = neovascularization.

were advanced into the femoral vein, and the laser fiber was withdrawn directly into the SFJ and then activated after flushing in the tumescent solution (Fig 2A–2C).

The second recurring pattern is short stumps from the GSV, which are directly connected to major tributaries, e.g. to the anterior accessory great saphenous vein. These can be cannulated below the SFJ, and the laser fiber can be advanced to the SFJ through such relatively straight vein sections (Fig 2D). The ablation then took place after flushing in the tumescence at 80–120 joules per cm.

## Subgroups

With regard to the technique of puncture, cannulation and placement of the laser fiber, two subgroups were compared: Procedures in which a direct puncture and cannulation of the stump had to be carried out; these cases were summarized in subgroup 1. Contrastingly, treatments in which a suitable straight vein segment was punctured and cannulated from the distal parts were outlined in subgroup 2.

## Endpoints

The treatment goal was defined as flush occlusion at the upper refluxing point at the level of the SFJ (Fig 3) and was determined by duplex ultrasound during follow up. The technical success was achieved when provocation by distal calf compression did not produce any pathological reflux in the current area of recurrent varicose veins at the level of the groin. The patients were recommended to take this follow-up visit after 10 days.

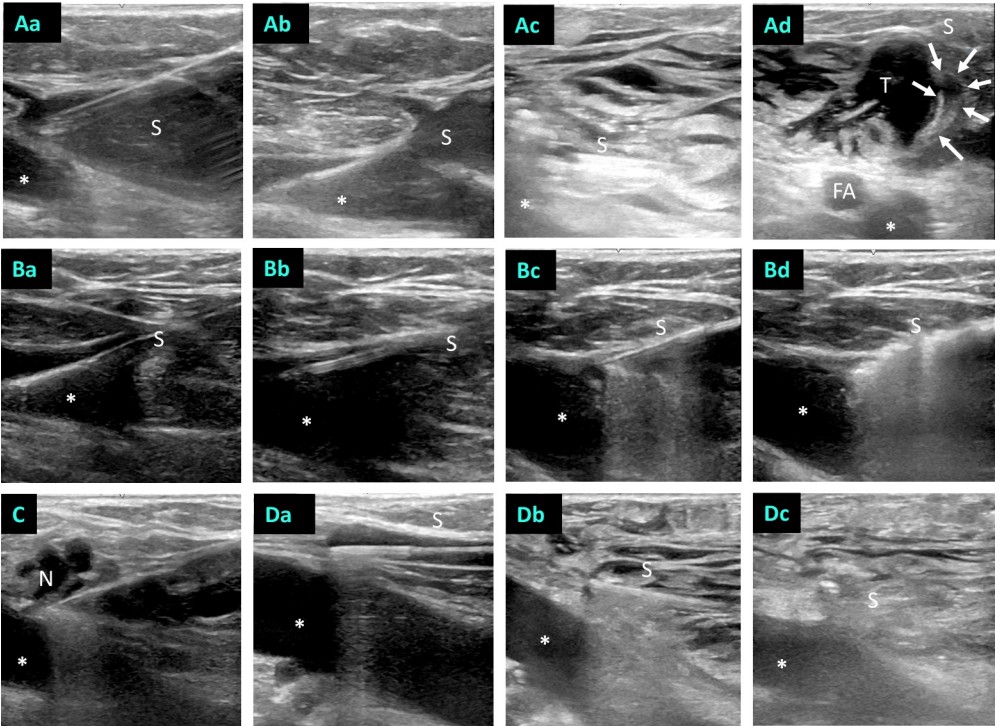

**Fig 2. Procedure of EVLA in SFJ recurrence.** (A) Direct puncture of a large stump (Aa) and placement of the guidewire in the femoral vein (Ab). (Ac) View after catheter placement and infusion of tumescent solution. (Ad) In transversal plane targeted infusion of tumescence solution (T) to increase the distance between stump (S; arrows) and the femoral artery (FA). * = femoral vein. (Ba-Bc) Another case to illustrate the positioning of the laser catheter after direct puncture and cannulation according to Seldinger through the stump (S). (Bd) Heat reaction after activating the laser energy. * = femoral vein. (C) An ultrasound image illustrating a laser fiber positioning throughout neovascularization (N). * = femoral vein. (D) An example illustrating the placement of the laser fiber by an intact, straight tributary of the GSV coming from distally. (Da) The catheter tip is exactly placed at the former sapheno-femoral junction. (Db) Tumescence solution is injected around the vein before ablation. (Dc) Thermal reaction after activation of the laser. * = femoral vein; S = stump.

## Statistics

A descriptive exploration of the entire sample was performed on basis of the treated limbs. These calculations, as well as the data management were conducted with Excel (Microsoft, Redmond, WA). In addition, the parameters collected were compared in the subgroups defined above, which differed in the method of cannulation and placement of the laser fiber.

Categorical baseline variables such as gender, American Society of Anesthesiologists (ASA)- Grade, Clinical-, Etiology-, Anatomy- and Pathophysiology (CEAP)-Grade, frequency of thrombophlebitis, type of pre-treatment, pre-treatment in own center were given as frequencies or percentages. The treatment-associated, categorical parameters anesthesia, presence of neovascularization, co-treatment of additional segments by EVLA, additional sclerotherapy, the lasers used and the technical success and complication rate were also presented as frequencies or percentages. The differences of categorical data between the subgroups were calculated using two-tailed Fisher's exact test of independence, where a p-value of less than 0.05 was considered statistically significant.

Continuous variables, including age, body mass index (BMI), time elapsed since pre-treatment, length and diameter of treated veins, and transmitted energy were tested for normality with Shapiro-Wilk test. Normally distributed parameters were presented as mean values with

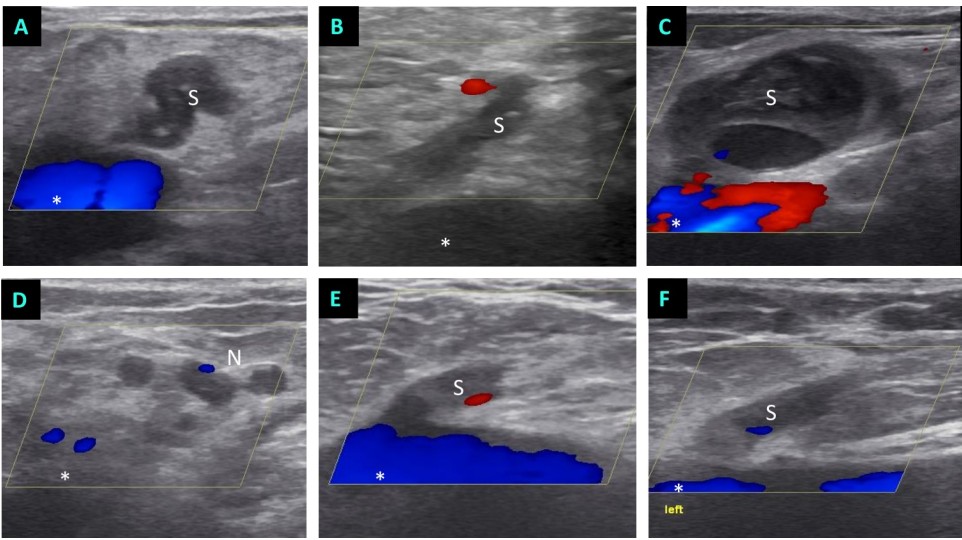

**Fig 3. Postoperative duplex ultrasound images.** Technical success is defined by flush occlusion of the (former) SFJ. Different EVLA-treated SFJ recurrences from subgroup 1 are shown in the upper row (A-C), which were addressed by direct puncture of a stump (S). (D) An example of neovascularization (N) that was also eliminated by direct puncture (subgroup 1). (E,F) Postoperative duplex ultrasound findings from two different patients from subgroup 2. * = femoral vein; S = stump, N = neovascularization.

standard deviations (SD). Parameters not normally distributed were expressed as median and range. Continuous data were compared by two-tailed Mann-Whitney U test, with significance

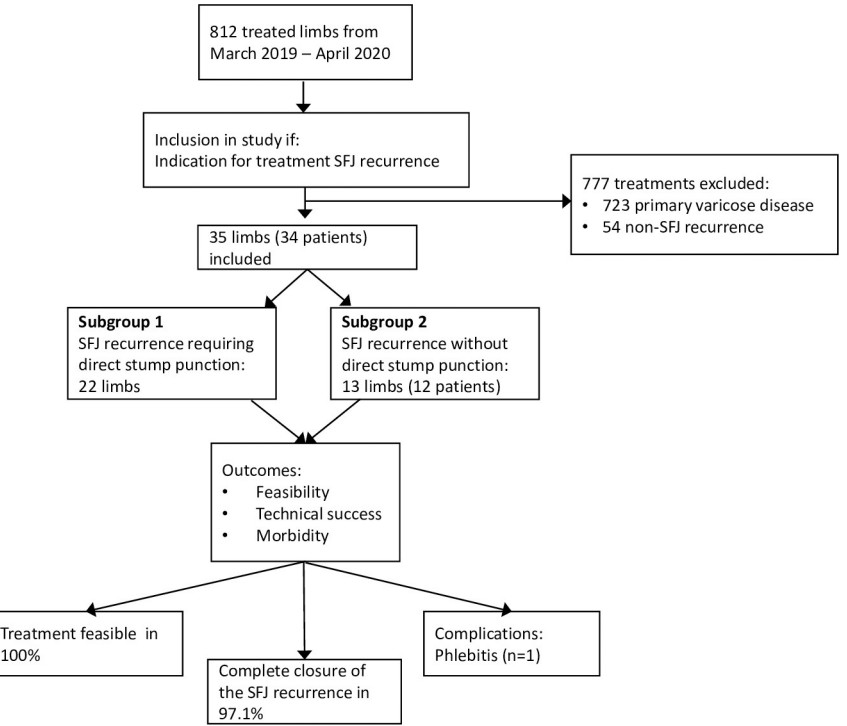

**Fig 4. Flow chart of study design.** SFJ, sapheno-femoral junction.

deemed at a p-value < 0.05. Statistical calculations were performed with GraphPad Prism (GraphPad Software Inc, San Diego, CA).

## Results

### Patients

In total, there were 812 limbs treated in 682 sessions. From these, 35 EVLA treatments in 34 patients were executed for the same site recurrences deriving from the SFJ (Fig 4). The mean age was 60.9 ± 13.4 years, and 22 patients were female. The subsequent analyzes and calculations of all parameters are based on the number of extremities treated.

There were 22 extremities in CEAP grade C2, 9 with grade C3, 3 with C4 and one with C6. Thrombophlebitis was diagnosed preoperatively in 3/35 (8.6%), all cases from subgroup 1. The median diameter of the vein to be treated at the upper insufficiency point was 7 mm (range: 5–21), with no significant differences between the subgroups. The baseline characteristics, as well as venous disease-specific characteristics, are presented in Table 1.

### Treatment

Due to the different morphologies of the recurrences, there were two distinct treatment modalities. First, cases with direct puncture and cannulation of a short-length stump or neovascularization (Fig 1A–1D). These treatments were summarized as subgroup 1 and comprised 22 treatments. Instead, procedures in which a puncture of a distal, straight vein segment and the advancement of the laser fiber to the SFJ was possible (Fig 1E and 1F). The latter were recapped to subgroup 2 and comprised 13 treatments (Table 1). As shown in Table 1, no differences could be detected in our cohort regarding the type of pre-treatment (operative versus endothermal). Moreover, the differences regarding demographic, and venous disease-specific parameters such as CEAP clinical grade and vein diameters did not differ significantly. There was also no difference in the type of anaesthesia between the groups.

The most striking difference between the subgroups was the underlying anatomy. Neovascularizations, as part of the venous tree to be ablated, were observed significantly more frequently in subgroup 1, affecting the treatment strategy in 81.8%. In contrast, these were apparent in only 15.4% in subgroup 2 (p < 0.001). As a result of the predominance of neovascularization-like phenotype in subgroup 1, there was a significantly shorter average vein length to be treated in the area of the SFJ, compared to subgroup 2 (Table 1). In the subgroup 1, 4/22 instances required direct stump puncture without the presence of apparent neovascularizations. In 3/22 cases, there was large diameter, proximally recanalized great saphenous vein after endothermal treatment. 1/22 the procedure was done for a varicose transformed, native anterior accessory great saphenous vein, which could only be punctured directly in the groin region due to its anatomy. In 86.4% of subgroup 1, the second ablation of a persistent trunk vein segment or a major varicose side branch via a separate puncture and cannulation was necessary, while in subgroup 2 it was only carried out in 30.8% (p = 0.002). Given the necessity of ablation of the short stump at the SFJ and the then regular thermal switching off of an additional vein segment, the complexity of laser ablation was greater in subgroup 1. No differences were found with regard to the use of the various radial lasers, whereby the 1940 nm radial laser was used less frequently but was only available for the last quarter of the investigation period. In addition, the diameter of the fiber used, on which the access system was dependent, was not different. Simultaneous foam sclerotherapy using ethoxysclerol of superficial varices below the middle of the thigh was performed in 48.6% of the patients, with no differences between the groups being observed.

**Table 1. Baseline and venous disease characteristics, treatment, and outcome parameters.**

| Parameter | Total limbs treated | Subgroup analysis | | p-value |
|---|---|---|---|---|
| | | Subgroup 1 | Subgroup 2 | |
| | | Direct stump puncture | Catheter placement from distal over a suitable vein segment | |
| | N = 35 | N = 22 | N = 13 | |
| Age, years, mean (SD) | 60.9 (13.4) | 64.7 (13.0) | 54.5 (12.0) | 0.097 |
| Male/female | 13/22 | 10/12 | 3/10 | 0.282 |
| BMI, kg/m2, mean (SD) | 26.8 (3.6) | 27.3 (3.0) | 26 (4.4) | 0.312 |
| ASA-Grade | | | | |
| Grade 1+2/Grade 3 | 32/3 | 20/2 | 12/1 | 1.000 |
| CEAP-clinical grade | | | | |
| C2/C3-6 | 22/13 | 11/11 | 11/2 | 0.070 |
| Thrombophlebitis, n (%) | 3 (8.6) | 3 (13.6) | 0 (0) | 0.279 |
| Type of pre-treatment | | | | |
| Surgical/Endothermal | 18/17 | 12/10 | 6/7 | 0.733 |
| Time from pre-treatment, Years, median (range) | 6 (1–26)[a] | 6 (1–26)[b] | 5.5 (3–25)[c] | 0.790 |
| Pre-treatment in own institution, n (%) | 15 (42.9) | 9 (40.9) | 6 (46.2) | 1.000 |
| Vein diameter, mm, median (range) | 7 (5–21) | 7 (5–21) | 7 (5–12) | 0.920 |
| General anaesthesia, n (%) | 27 (77.1) | 17 (77.3) | 10 (76.9) | 1.000 |
| Neovascularization affecting ablation strategy, n (%) | 20 (57.1) | 18 (81.8) | 2 (15.4) | < 0.001 |
| Length of the proximally treated vein segment, cm, median (range) | 2.5 (1–38) | 1.6 (1–3) | 16 (6–38) | < 0.001 |
| Total energy transferred to the proximal segment, J, median (range) | 304 (115–2754) | 262 (115–440) | 946 (405–2754) | 0.001 |
| Concomitant EVLA of additional vein segment, n (%) | 23 (65.7) | 19 (86.4) | 4 (30.8) | 0.002 |
| Length of additional vein segment, cm, median (range) | 13 (4–38) | 10 (4–36) | 18.5 (5–38) | 0.143 |
| Total energy transferred to additional vein segment, J, median (range) | 701 (107–3785) | 691 (198–1681) | 1140 (107–3785) | 0.372 |
| Radial laser wavelength | | | | |
| 1470 nm/1940 nm | 33/2 | 21/1 | 12/1 | 1.000 |
| Laser Fiber diameter | | | | |
| 1.8 mm/1.3 mm | 16/19 | 13/9 | 4/9 | 0.164 |
| Concomitant sclerotherapy, n (%) | 17 (48.6) | 11 (50) | 6 (46.2) | 1.000 |
| Time to postoperative control, days, median (range) | 10 (3–60) | 10 (4–29) | 11 (3–60) | 0.850 |
| Technical success, n (%) | 34 (97.1) | 21 (95.5) | 13 (100) | 1.000 |
| Minor complications, n (%) | 1 (2.9) | 0 (0) | 1 (7.7) | 0.371 |

ASA, American Society of Anesthesiologists; BMI, body mass index; CEAP, Clinical, Etiology, Anatomy and Pathophysiology; EVLA, endovenous laser ablation; Min, minimum; Max, maximum; SD, standard deviation.

[a]Data missing for 6 limbs

[b]Data missing for 5 limbs

[c]Data missing for 1 limb

## Feasibility

All procedures could be performed intra-operatively as planned, and there were no noticeable failures. In 10/35 treatments, additional diseased trunk veins at different sites were treated with EVLA. In one case it was the small saphenous vein of the same extremity, in the other situations the opposite side was affected. No intraoperative complications, such as bleeding or

anesthesia, were observed. The patients were able to start their usual daily activities the next day, including sporting activities. Compression stockings were not prescribed.

## Complications

Procedural related complications were observed in 1/35 cases. One individual from subgroup 2 developed a phlebitic reaction in the area of the superficial epigastric vein during the postoperative course, which improved quickly after anti-inflammatory medication and local cooling. There were no major complications, especially no thromboembolism, and no endovenous heat-induced thrombosis.

## Technical success rate

The technical success of duplex sonography was determined as part of the follow-up examination. The follow-up rate was 100%. Patients were advised to have this test done in about ten days, although there were outliers in both directions. The median time to postoperative checkup was 10 days (range: 3–60). The technical success, defined as thermally induced, flush closure of the upper insufficiency point (Fig 3), was given in 34/35 cases (97.1%). In one case from subgroup 1, there was only an inadequate closure sonographically, and further foam sclerotherapy was recommended for this patient.

## Discussion

From the present case series, laser ablation of recurrences of the same site from the former SFJ appears feasible, with comparable low postoperative morbidity to EVLA treatment for primary varicosis. This is supported on the one hand by the overall observed low complication rate and on the other hand, by the comparison of the subgroups. In subgroup 2, the puncture and cannulation of the vein to be treated proceeded from distal, and the insertion of the laser fiber and insertion through a relatively straight vein segment were carried out in the same way as with the primary treatment of incompetent GSV.

There is little data in the literature on the endothermal treatment of such SFJ recurrences. In an Italian study, the procedure was described in a manner comparable to that shown here [15]. Eight treatments of short stumps were performed on the SFJ with the 1.3 mm 1470 nm radial fiber. With a mean follow-up time of 8 months, only one recanalization of the treated stump was observed, and the average vein diameter at 10.2 mm was even higher than in our analysis. We utilize the thicker 1.8 mm radial fiber that closes potentially more effective large-caliber veins, usually for larger-caliber stumps. With the 1.8 mm fiber, however, access is made using a guide wire and introducer sheath, which means that the femoral vein is instrumented for short stumps (Fig 2A and 2B). As another difference, the authors describe the use of foam sclerotherapy in the stump area. Foam sclerotherapy was also used in our cases, but only downstream of the thermally ablated segments, below the middle of the thigh. This was because through the endothermal treated vein, and the presence of tumescence control of foam distribution was difficult. Thereby the risk of accumulation of ethoxysclerol in the femoral vein was considered too great.

Another Dutch retrospective analysis published in 2009 describes the treatment of recurrent varicose veins using EVLA (n = 67) in comparison to surgery (n = 149) for recurrent varicose veins [16]. Only cases in which there were no tortuous veins were explicitly treated with laser. It is described that the tip of the 600 μm bare fiber utilized was placed within the introductory sheath and 20 mm away from the SFJ. Although the comparability of this study with the technique described here appears to be limited, it is interesting to note that clinically relevant recurrences occurred less frequently 25 weeks after treatment after EVLA (19%) than

after surgery (29%). A third, English study described the use of the EVLA for recurrent varicose veins, but only in situations in which there was a relatively straight stem vein segment of at least 10 cm in length, while short stumps or neovascularizations were not considered suitable [17]. In these cases, the authors preferred the EVLA to the operation. Overall, the amount of literature on recurrence treatment by endothermal procedures appears surprisingly small because of the widespread use of these techniques, including their implementation in national guidelines, and gives hope that further studies on this topic will appear.

The sonographic finding and the histological result can be discrepant in the SFJ recurrence [22]. For this reason, we only selected a morphological description in our study for subgroup definition from which it cannot always be established whether shortcomings in pre-treatment play a role. Another reason was that the introduction of endovenous techniques made the classification of recurrences even more complex. In our study, for example, 13 cases that formed subgroup 2 were treated in which an intact, relatively straight and probably original trunk vein or a large branch vein was present after the pre-treatment (Table 1). According to the general definition, there would be a technical or tactical error, since the high ligation and stripping, which includes switching off and separating the tributaries of the GSV, should prevent from such recurrence pattern. In contrast, this form of recurrence after endovenous treatment is not a technical error, since the simultaneous treatment of tributaries that are not refluxive and widen at the time of GSV treatment is usually not intended [23].

Open surgical treatment for SFJ relapses can be problematic. Postoperative morbidity was reported in up to 40% of cases, including many lymph fistulas, wound infections, and hematomas [19,20]. Also, special suturing or barrier techniques that invert the femoral vein endothelium did not improve long-term result [20,24]. Compared to the surgical procedures, EVLA presumably offers an advantage in terms of morbidity. There are data on reduced, compared to surgery, formation of neovascularizations after primary treatment with endothermal ablations [25,26]. Further studies should clarify whether this also applies to the treatment of SFJ recurrences.

The limitations of this study, which have to be discussed, result on the one hand from the study design. There is only a retrospective, consecutive case series with which no hypothesis is checked. Due to the subgroup analysis, which refers to a different, defined technical procedure for ablation, an analytical approach is given. Together, our observations raise the hypothesis that, despite greater technical complexity, the treatment of the SFJ recurrence with EVLA is comparable to the EVLA for primary varicose disease in terms of peri-interventional morbidity. Another limitation arises from the lack of a long-term follow-up. However, we felt that the duplex ultrasound 1–2 weeks after an EVLA procedure carries sufficiently diagnostic information to estimate the initial technical result of occlusion.

## Conclusion

EVLA of SFJ recurrences appears feasible, also in the presence of complex anatomical situations like short stumps or tortuous, neovascularization-like vein structures. It would now be interesting to compare the endovascular treatment technique with the open surgical technique in terms of feasibility and morbidity in the context of a prospective study.

## Supporting information

**S1 Table. Database.**
(XLSX)

## Author Contributions

**Conceptualization:** Lars Müller.

**Data curation:** Lars Müller.

**Formal analysis:** Lars Müller.

**Investigation:** Lars Müller.

**Methodology:** Lars Müller.

**Project administration:** Lars Müller, Jens Alm.

**Resources:** Lars Müller, Jens Alm.

**Supervision:** Jens Alm.

**Validation:** Lars Müller, Jens Alm.

**Visualization:** Lars Müller.

**Writing – original draft:** Lars Müller.

**Writing – review & editing:** Lars Müller, Jens Alm.

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
