## [Decision Letter · Decision Letter 0]

10 Jun 2020

PONE-D-20-14421

Feasibility and technique of endovenous laser ablation (EVLA) of recurrent varicose veins deriving from the sapheno-femoral junction – a case series of 35 consecutive procedures

PLOS ONE

Dear Dr. Müller,

Thank you for submitting your manuscript to PLOS ONE. After careful consideration, we feel that it has merit but does not fully meet PLOS ONE’s publication criteria as it currently stands. Therefore, we invite you to submit a revised version of the manuscript that addresses the points raised during the review process.

Please address the reviewer's concerns and make revisions accordingly. 

We look forward to receiving your revised manuscript.

Kind regards,

Academic Editor

PLOS ONE

Journal Requirements:

2. In your ethics statement in the manuscript and in the online submission form, please provide additional information about the patient records used in your retrospective study. Specifically, please ensure that you have discussed whether all data were fully anonymized before you accessed them.

3. To comply with PLOS ONE submission guidelines, in your Methods section, please provide additional information regarding your statistical analyses. For more information on PLOS ONE's expectations for statistical reporting, please see https://journals.plos.org/plosone/s/submission-guidelines.#loc-statistical-reporting.

Reviewers' comments:

Reviewer's Responses to Questions

**Comments to the Author**

1. Is the manuscript technically sound, and do the data support the conclusions?

Reviewer #1: Partly

2. Has the statistical analysis been performed appropriately and rigorously? 

Reviewer #1: N/A

3. Have the authors made all data underlying the findings in their manuscript fully available?

Reviewer #1: Yes

4. Is the manuscript presented in an intelligible fashion and written in standard English?

Reviewer #1: Yes

5. Review Comments to the Author

Reviewer #1: The presenting study is a case series.

(1) Case series (and case reports) per definition do not provide any comparison groups. With the introduction of the subgroup analysis of a relatively small number of cases, the reader might confuse the design with the retrospective cohort study (where the control group is present and advisable). I suggest making sure that there is a clear notion about the study design (case series) in the text body - as the authors acknowledged in the title.

(2) The major pitfall for the clinician is to conclude about relationships from case series. The inferences about morbidity cannot be made, based on relatively small numbered case series. I suggest rewriting the conclusion.

The most interesting question raised here would be to address the possible prospective observational study comparing open surgery (OS) to described EVLA technique in terms of morbidity and technical success rate.

6. PLOS authors have the option to publish the peer review history of their article (what does this mean?). If published, this will include your full peer review and any attached files.

Reviewer #1: No

---

## [Author Response · Author response to Decision Letter 0]

15 Jun 2020

Dr. Lars Müller

Department of Vascular Surgery

Dermatologikum Hamburg

Stephansplatz 5

20354 Hamburg

Germany

To

Dr. Robert Jeenchen Chen

Academic Editor

PLOS ONE

1160 Battery Street

Koshland Building East, Suite 225

San Francisco, CA 94111

United States

 Hamburg, 15.6.2020

Dear Dr. Chen, 

I would like to thank you, also on behalf of my co-author Dr. Alm, for the constructive review process. I also thank the reviewers for their time and effort with our manuscript.

We have revised our manuscript in several respects on the basis of your criticism and suggestions and would now like to explain them to you. For this purpose, we will precede the comments (in bold letters) made by you and by the reviewer and then display our explanations and the respective changes in the manuscript:

Reviewer #1: The presenting study is a case series.

(1) Case series (and case reports) per definition do not provide any comparison groups. With the introduction of the subgroup analysis of a relatively small number of cases, the reader might confuse the design with the retrospective cohort study (where the control group is present and advisable). I suggest making sure that there is a clear notion about the study design (case series) in the text body - as the authors acknowledged in the title.

We agree. We have now referred to the design of the study (case series) at several places in the manuscript. We have also adjusted or scaled down the statements that can be drawn from a case series at several points.

-Abstract: Conclusions Statement (page 3, line 50)

-Introduction: Page 4, line 84

-Discussion: page 15, line 296/297 

-Discussion: page 18, line 356 (Observation instead of results)

-Discussion: page 18, line 361

We have also added more specific information to the abstract in order to make it easier to understand what exactly was examined (changes on page 2, lines 28-44)

(2) The major pitfall for the clinician is to conclude about relationships from case series. The inferences about morbidity cannot be made, based on relatively small numbered case series. I suggest rewriting the conclusion.

The most interesting question raised here would be to address the possible prospective observational study comparing open surgery (OS) to described EVLA technique in terms of morbidity and technical success rate.

We can only agree with this comment, and we have immediately adopted this argument in our conclusion (page 18, lines 366-368).

We have revised the names of the image files.

2. In your ethics statement in the manuscript and in the online submission form, please provide additional information about the patient records used in your retrospective study. Specifically, please ensure that you have discussed whether all data were fully anonymized before you accessed them.

We have reformulated the Ethics Statement, and also the previous chapter Study Population (page 5, lines 94-108)

3. To comply with PLOS ONE submission guidelines, in your Methods section, please provide additional information regarding your statistical analyses.

The chapter on statistics (page 9, line 194 - page 10, line 213) has been rewritten and the text has been changed accordingly:

-Page 10, line 224: Median and Range instead of Average and Standard Deviation.

-Table 1: Median and range is presented for all data which is not normally distributed.

-Page 15, line 289-290: The median follow-up time is now given instead of the average.

4. Please include captions for your Supporting Information files at the end of your manuscript, and update any in-text citations to match accordingly.

This caption at the end of the manuscript was included.

Other small changes that should improve the understanding of the text from our point of view: 

-Page 4, line 76

-page 6, line 120-121

-Page 7, line 151

-Page 7, line 160

-The chapter endpoints slightly shortened and rearranged (page 8. lines 178-184)

-Page 13, line 236

-Page 15, line 302

-page 17, line 340, 341

-Supplemental data table (S1_Table): Some corrections have been made to the column labels.

I look forward to hearing from you regarding our submission. We will be happy to continue to answer any further suggestions or queries or to make any necessary changes.

Yours sincerely, 

Dr. Lars Müller

---

## [Decision Letter · Decision Letter 1]

22 Jun 2020

Feasibility and technique of endovenous laser ablation (EVLA) of recurrent varicose veins deriving from the sapheno-femoral junction – a case series of 35 consecutive procedures

PONE-D-20-14421R1

Dear Dr. Müller,

We’re pleased to inform you that your manuscript has been judged scientifically suitable for publication and will be formally accepted for publication once it meets all outstanding technical requirements.

Kind regards,

Academic Editor

PLOS ONE

Additional Editor Comments (optional):

Reviewers' comments:

Reviewer's Responses to Questions

**Comments to the Author**

1. If the authors have adequately addressed your comments raised in a previous round of review and you feel that this manuscript is now acceptable for publication, you may indicate that here to bypass the “Comments to the Author” section, enter your conflict of interest statement in the “Confidential to Editor” section, and submit your "Accept" recommendation.

Reviewer #1: All comments have been addressed

Reviewer #2: All comments have been addressed

2. Is the manuscript technically sound, and do the data support the conclusions?

Reviewer #1: Yes

Reviewer #2: Yes

3. Has the statistical analysis been performed appropriately and rigorously? 

Reviewer #1: Yes

Reviewer #2: Yes

4. Have the authors made all data underlying the findings in their manuscript fully available?

Reviewer #1: Yes

Reviewer #2: Yes

5. Is the manuscript presented in an intelligible fashion and written in standard English?

Reviewer #1: Yes

Reviewer #2: Yes

6. Review Comments to the Author

Reviewer #1: All my comments have been addresed. The methodological constrains of the case series have been adequately emphasized.

Reviewer #2: The manuscript is now acceptable for pubblication, the author have adequately addressed reviewer's comments.

7. PLOS authors have the option to publish the peer review history of their article (what does this mean?). If published, this will include your full peer review and any attached files.

Reviewer #1: No

Reviewer #2: Yes: GIOVANNI DE CARIDI

---

## [Editor Report · Acceptance letter]

24 Jun 2020

PONE-D-20-14421R1 

Feasibility and technique of endovenous laser ablation (EVLA) of recurrent varicose veins deriving from the sapheno-femoral junction – a case series of 35 consecutive procedures 

Dear Dr. Müller:

I'm pleased to inform you that your manuscript has been deemed suitable for publication in PLOS ONE. Congratulations! Your manuscript is now with our production department. 

Kind regards, 

on behalf of

Dr. Robert Jeenchen Chen 

Academic Editor

PLOS ONE